# Dispositional and situational personal features and acute post-collision head and neck pain: Double mediation of pain catastrophizing and pain sensitivity

**Michal Granot** [1]☯ *, **Einav Srulovici** [1]☯, **Yelena Granovsky** [2], **David Yarnitsky** [2,3], **Pora Kuperman** [3]

**1** Department of Nursing, University of Haifa, Haifa, Israel, **2** Faculty of Medicine, Technion- Israel Institute of Technology, Haifa, Israel, **3** Department of Neurology, Rambam Health Care Campus, Haifa, Israel

☯ These authors contributed equally to this work.
* granot@research.haifa.ac.il

**Data Availability Statement:** All relevant data are within the paper and its Supporting Information files.

## Abstract

Pain variability can be partially attributed to psycho-cognitive features involved in its processing. However, accumulating research suggests that simple linear correlation between situational and dispositional factors may not be sufficiently explanatory, with some positing a role for mediating influences. In addition, acute pain processing studies generally focus on a post-operative model with less attention provided to post-traumatic injury. As such, this study aimed to investigate a more comprehensive pain processing model that included direct and indirect associations between acute pain intensity in the head and neck, pain catastrophizing (using pain catastrophizing scale (PCS)), and pain sensitivity (using the pain sensitivity questionnaire (PSQ)), among 239 patients with post-motor vehicle collision pain. The effect of personality traits (using Ten Items Personality Inventory (TIPI)) and emotional status (using Hospital Anxiety and Depression Scale (HADS) and Perceived Stress Scale (PSS)) on that model was examined as well. To this end, three Structural Equation Modeling (SEM) analyses were conducted. Overall, the data had good fit to all the models, with only PSQ found to have a direct correlation with acute pain intensity. The SEM analyses conversely revealed several mediations. Specifically, that: first, PSQ fully mediated the relationship between PCS and pain intensity; second, PCS and PSQ together fully mediated the relationship between conscientiousness (personality trait) and pain intensity; and finally, emotional status had direct and indirect links with PSQ and pain intensity. In conclusion, these models suggest that during the acute post-collision phase, pain sensitivity intermediates between emotional states and personality traits, partially via elevated pain catastrophizing thoughts.

**Funding:** D.Y. received grant W81XWH-15-1-0603 from the US Department of Defense, Health Affairs Office. The funders had no role in study design, data collection and analysis, decision to publish, or preparation of the manuscript.

**Competing interests:** D.Y, holds equity in BrainsGate Ltd. And Theranica Ltd. All other authors report no relevant disclosures or potential conflicts of interest. This does not alter our adherence to PLOS ONE policies on sharing data and materials.

# 1. Introduction

Nearly half of individuals who suffer acute pain due to injury resultant from motor vehicle collisions will go on to report chronic pain [1, 2]. As such the need to explore potential predictors which affect the transition from acute to chronic pain is obvious. However, while much research addresses the correlation between personal traits and emotional states with the psychology of pain among chronic pain patients, affective processing is less well-understood in the context of acute pain intensity, such as that of post-collision. Given that, enhanced acute pain intensity is the main predictor for chronic pain, addressing the dynamic role of psycho-cognitive features as well as processes that underlie sensory and affective responses to nociceptive input during the early acute pain phase may shed more light on the observed variability in the magnitude of the pain experience [3–6].

A traditionally key element in determining variability in pain intensity is pain catastrophizing (PC), which was conceptualized as the tendency to overestimate the severity and consequence of pain [7]. To quantify PC, the Pain Catastrophizing Scale (PCS) was developed to incorporate feature of rumination, magnification, and helplessness toward pain, assumed to attain either situational or dispositional elements [7, 8]. While the latter is considered as a stable characteristic, the former might be manifested in more flexible manner, such that under particular circumstances, the expression of situational features may be altered after an exposure to a demanding situation in either a short- or long-term manner. Although, the link between PCS and pain can be changed over time—highlighting its situational characteristic, previous studies have proposed a reciprocal relationship between situational and dispositional PC. Namely, dispositional PC can be linked with pain intensity in the case that the nociceptive stimulus or the painful event evokes enhanced situational PC and vice versa [8–10].

Indeed, the PCS is a generally well-accepted predictor of negative pain-related outcomes and enhanced pain experience in both acute and chronic pain conditions [11–13]. Nevertheless, recent studies reported that PCS ratings were not associated with acute pain intensity [14–17]. Interestingly, higher pain sensitivity ratings, as obtained by the Pain Sensitivity Questionnaire (PSQ), which assume to depict perceived or imagined response to various everyday pain situations, were found to be directly associated with augmented intensity of acute pain experience [17]. Thus, PCS and PSQ warrant consideration, as either or both may serve either directly or indirectly as indicators of psycho-cognitive pain processing. In that they reflect the manner in which cognitive representation, memories and imagination toward pain shape its experience.

Two conceptual frameworks may be relevant to attain a broader understanding about the relationships of pain catastrophizing and pain sensitivity as well as personal traits and emotional states. First, the fear-avoidance model [18] indicates that an individual with catastrophic thoughts toward pain following injury will tend to avoid activities due to the manner in which the neuromatrix of nociceptive modulation processing enhances pain sensitivity which then determines pain experience. Accordingly, higher PC will directly augment pain sensitivity as expressed by enhanced pain intensity ratings. Based on this concept we hypothesized that:

*Hypothesis 1*: *Higher pain catastrophizing ratings will be associated with higher pain sensitivity, which in turn will be associated with higher acute pain intensity.*

The second theoretical framework is the disposition and adjustment to chronic pain model [19] which suggest that personality traits and emotional states (i.e., mental states) might shape PC based on the individual vulnerability (e.g., stress, anxiety) and resource characteristics available (e.g., extraversion, consciousness) toward pain. While an individual's vulnerability characteristics shape PC through activation of physiological mechanisms evoked in response

to nociceptive stimulus, individual resource characteristics shape adaptive coping mechanism recruited to allow them to manage their response to pain. Thus, each individual is located in a different position on the continuum of pain modulation profiles [20] due to the specific amalgamation of 'stable-dispositional' and 'temporary-situational' personal characteristics [21, 22]. This notion was supported by recent publications highlighting the role of mediation and/or moderation effects [23]. For example, PCS was found as mediator between personality traits and pain intensity [24–26]. Specifically, higher scores on agreeableness, extraversion, open to experiences, and conscientiousness (e.g., resource characteristics) were associated with lower PCS and higher scores on neuroticism (e.g., vulnerability characteristic) were associated with higher PCS [27]. The role of PCS, as an independent variable in Hypothesis 1, might also be mediated through pain sensitivity.

*Hypothesis 2a*: *Higher scores on personality traits considered 'resources' according to the disposition and adjustment to chronic pain model will be associated with lower pain catastrophizing ratings, which will in turn be associated with lower pain sensitivity, resulting in lower acute pain intensity.*

*Hypothesis 2b*: *Higher scores on personality traits considered 'vulnerabilities' according to the disposition and adjustment to chronic pain model will be associated with higher pain catastrophizing ratings, which in turn will be associated with higher pain sensitivity, leading to higher acute pain intensity.*

Furthermore, emotional states associated with the post-injury circumstance, such as stress, depression, and anxiety, have been linked with PC [28, 29]. With this, mounting research has failed to find direct correlations between state anxiety and depression, and pain in acute post-operative patients [30–32], and emotional elements have been found to be only mildly related to PSQ [33, 34]. Thus, it is possible that the three are interrelated, where emotional state elements (i.e., emotional status) might be indirectly related to pain intensity through PC, and as suggested in Hypothesis 1, through pain sensitivity.

*Hypothesis 3*: *A heightened post-collision emotional status will be linked to higher PC ratings, followed by higher pain sensitivity and consequently higher acute pain ratings.*

To explore these hypotheses, we chose to apply structural equation modeling (SEM), an advanced statistical approach, which allows for the simultaneous examination of direct and indirect relationships among latent dispositional and situational characteristics, and acute post-traumatic pain intensity. Specifically, the current study aimed to test three models (Fig 1). The first aimed to explore the mediating role of pain sensitivity in the link between PC and acute post-traumatic pain intensity; the second aimed to investigate how the first model is affected by personality traits; and finally, the third aimed to examine how the first model is affected by emotional status.

## 2. Methods

### 2.1. Study design

Patients are part of a larger prospective non-interventional study where initial data was collected between March 2016 and December 2019. A session was scheduled within 72 hours post-injury for MRI, clinical, psychophysical (i.e., experimentally induced pain assessment), psycho-cognitive and neurophysiological assessment which was completed at the testing site. Additionally, blood was drawn for genetics and patients' demographic and clinical baseline assessments (i.e., socio-demographic information, self-reported pain levels, areas of post-

Model 1. The mediating role of PSQ on the link between PCS and pain intensity

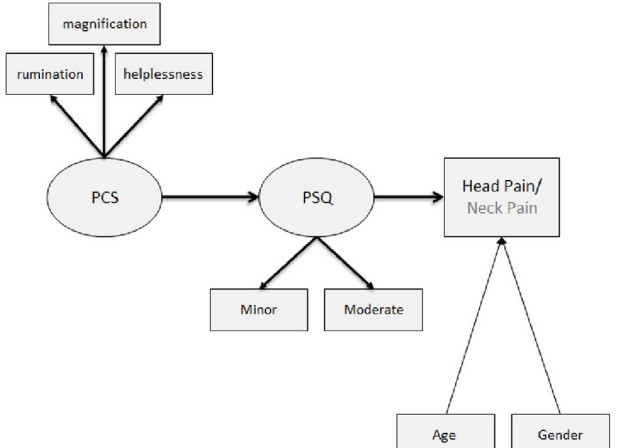

Model 2. The med-med model of personality traits on pain intensity

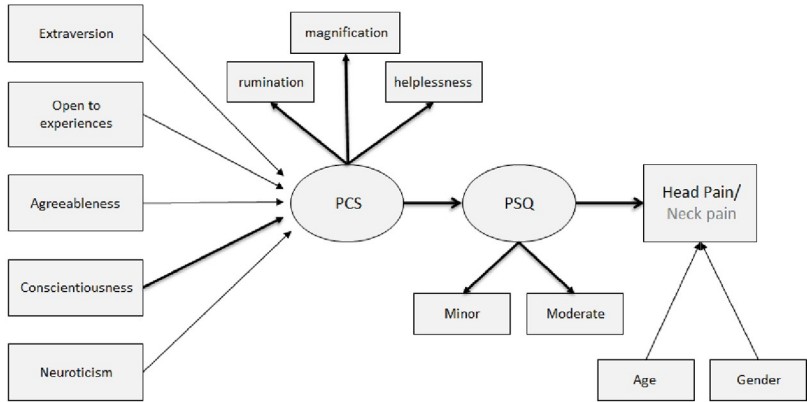

Model 3. The med-med model of mental state on pain intensity

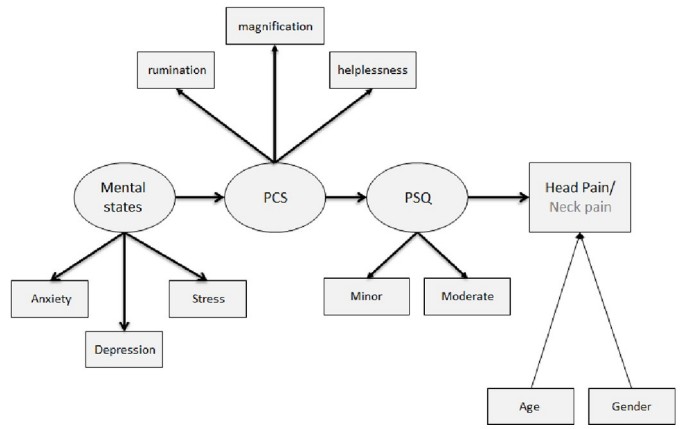

**Fig 1. Outline of proposed SEM analyses.**

accident body pain, and use of analgesics) were obtained. For more information on the full study protocol please see our previous work [17, 35, 36].

## 2.2. Study population

**2.2.1. Participants.** Patients were recruited when visiting the Rambam HealthCare Campus Emergency Room in Haifa, Israel. Inclusion criteria: road accident up to 24 hours before ER arrival; direct or indirect head and neck injury with reports of pain, Glasgow coma scale (GCS) 13–15 with no subsequent decline; no traumatic findings in computed tomography (CT) if performed; age 18–70, both males and females. Exclusion criteria: lack of ability to communicate in Hebrew; other major bodily injuries from the accident; prior chronic head/neck pain that requires regular treatment; neurological disease that might affect testing ability or interpretation such as neurodegenerative diseases; any head and neck injury in past year; any pain condition that requires daily dose of pain medication. The institutional review board of Rambam Health Care Campus approved the study protocol in accordance with The International Helsinki Declaration (No. 0601–14). Written informed consent was obtained from each participant in the presence of a certified physician prior to any data collection or assessment.

## 2.3. Measures

**2.3.1. Dependent measure.** *2.3.1.1. Acute pain intensity*. Was assessed via a Visual Analog Scale (VAS) scale of 0–100 (0 represents 'no pain', 100 represents 'worst pain imaginable') for the following parameters as it related to the preceding 24h: mean pain scores in the neck and in the head. Participants provided this rating via a custom made smart-phone application. As the cohort was comprised of post-collision individuals, and the study focused on the trajectory of individuals with initial area-of-injury pain (i.e., head and neck), the mean rating for both were considered primary outcome measures in the current study.

**2.3.2. Independent measures.** *2.3.2.1. The Ten Items Personality Inventory (TIPI)*. [37, 38], is a questionnaire used to assess the 5 dimensions of the Five Factor Model (FFM) of personality (neuroticism, extraversion, conscientiousness, open to experiences, agreeableness). Each factor is independent and includes pairs of personality trait descriptors rated on a 7-point Likert scale (1 = strongly agree to 7 = strongly disagree). Sample items include the following: "I see myself as anxious or easily upset" and "I see myself as dependable or self-disciplined." Scores are averaged for each factor, where each personality factor represents a continuum of trait characteristic that ranges between two anchors in which the middle point represents the baseline [39].

*2.3.2.2. Emotional status*. This latent variable contained two measures that represent emotional status. Higher scores are considered as higher emotional distress (i.e., high stress, anxiety, and depression levels).

*2.3.2.3. Perceived Stress Scale (PSS)*. [40]- a self-report 10-item questionnaire devised to measure the perception of stress. It is a measure of the degree to which situations which occurred within the last month are appraised as stressful. The items are designed to assess how unpredictable ('how often have you been upset because of something that happened unexpectedly'), uncontrollable ('how often have you felt that you were unable to control the important things in your life?') and overloaded ('how often have you felt that you were on top of things?') the subjects find their lives to be. The items are rated from 0 ('never') to 4 ('very often'), 6 items are worded negative and 4 are positive. The subjects were instructed to relate to the accident as part of their last month. Cronbach alpha in the current study was 0.783.

*2.3.2.4. Hospital Anxiety and Depression Scale (HADS).* [41]- a self-report 14-item questionnaire devised to be used to measure anxiety and depression in individuals with physical health problems. The questionnaire focuses on non-physical symptoms so that it can be used to diagnose depression in people with significant physical ill health. The items are rated from 0 (negative response) to 3 (very positive response). Seven of the items relate to anxiety, and 7 depression, and as such HADS provides two scores. The range for each subscale is 0–21 points, with higher scores indicating more symptoms of anxiety and depression. Cronbach alpha in the current study was 0.866 for anxiety and 0.727 for depression.

**2.3.3. Mediating measures.** Patients filled out the following questionnaires, using the Hebrew validated version [11, 42]:

*2.3.3.1. Pain Catastrophizing Scale (PCS).* [7]—a self-report 13-item questionnaire providing ratings based on painful life situations. Catastrophizing is conceptualized by cognitions related to the inability to tolerate painful situations, thinking pain is unbearable, or ruminating on the worst possible outcomes from the pain which is being experienced. As such, the instrument represents the three components of pain catastrophizing: rumination (e.g., "I can't seem to keep it out of my mind"); magnification (e.g., "I wonder whether something serious may happen"); and helplessness (e.g., "There is nothing I can do to reduce the intensity of pain"). Participants are asked to rate each statement on a 5-point Likert scale ranging from 0 ('not at all') to 4 ('always'). The PCS provides a total score and three sub-scores. The three sub-scores of PCS were used for analysis. Patients were not directed to focus on any particular pain event they experienced in the past [7]. Cronbach alpha in the current study was 0.896.

*2.3.3.2. Pain Sensitivity Questionnaire (PSQ).* [43]—a self-report 17-item questionnaire, based on pain intensity ratings of imagined painful daily life situations touching on various somatosensory sub-modalities. The items are rated from 0 ('not painful at all') to 10 ('worst pain imaginable'), and span various thermal, chemical, and mechanical pain modalities, noxious intensities and body sites. 14 items relate to situations that are painful for the majority of persons. For example: "Imagine you trap your finger in a drawer" and "Imagine you pick up a hot pot by inadvertently grabbing its equally hot handles". The remaining 3 items describe normally non-painful situations. For example: "Taking a warm shower". The latter are interspersed in order to serve as non-painful sensory references for the subjects. The PSQ provides a total score and two sub-scores of minor and moderate. The PSQ total score was calculated as the average rating of all but the three non-painful items. A higher PSQ score indicates higher pain sensitivity. Cronbach alpha in the current study was 0.932.

**2.3.4 Control variables.** As suggested in previous studies [e.g., 44], we collected participants' age and gender to control for possible effects on pain perception.

## 2.4. Statistical analysis

First, descriptive statistics (mean and standard deviation, and proportions, as appropriate) and correlations were conducted for all study's variables. Second, SEM mediation analyses were conducted. The SEM models were based on recent recommendations for mediation examination [45], where the independent variable can be linked to the dependent variable only through the mediator. Following Bowen and Guo's [46] recommendation, alternative models that are based on the conservative recommendation of Baron and Kenny (1986), where the independent variable must have a direct link with the dependent variable [47], were also conducted. Specifically, six SEM analyses that simultaneously examined the direct and indirect relationships among personality traits, emotional status, PCS, PSQ, and head and neck pain intensity ratings were performed in addition to the inclusion of known control variables (i.e., gender and age). This analysis was comprised of three primary SEM analyses and an alternative model

for each one, which examined a direct path between the independent variables and head and neck pain intensity ratings to emphasize best fit.

The fit of the data to the model was assessed, as accepted in the field of SEM [46], using a maximum likelihood estimator and several fit indices: the chi-square test ($\chi2$), the comparative fit index (CFI), the Tucker-Lewis index (TLI), and the root mean square error of approximation (RMSEA). The data were assumed to fit the model when the non-significant chi-square test [48]; the obtained CFI and TLI were greater than 0.90; and the obtained RMSEA was lower than 0.06 [49]. In order to determine whether the initial model or the alternative model is the preferred one, the difference in chi-squares and degrees of freedom between the initial model and the alternative model was examined using the chi square distribution table. A non-significant difference between the models indicated that the simpler model (with more degrees of freedom) is the preferred model. However, a significant difference between the models indicates that the less parsimonious model is the preferred one [46]. A minimum sample size of 138 for the first model, 200 for the second model, and 156 for the third model was needed to detect a small effect size with a power of 80% under alpha .05 [50, 51]. Descriptive statistics and bivariate analyses were performed using SPSS software version 25, and models were tested using SEM using IBM SPSS AMOS version 25.0. Significance was set at $p < 0.05$.

## 3. Results

### 3.1. Description of study cohort

A total of 239 acute post-collision patients were included in this study. About half were male (n = 134, 56.1%), aged 37.6 (±12.4) years on average (range 18–67 years old). The mean pain intensity ratings were 47.65 (±27.66) and 52.49 (±28.13) for head and neck pain, respectively. Descriptive statistics of study variables are presented in Table 1.

**Table 1. Descriptive statistics of study variables.**

| Study variables | Mean | St. Deviation |
|---|---|---|
| Head pain | 47.65 | 27.66 |
| Neck pain | 52.49 | 28.13 |
| PSQ–pain sensitivity | | |
| PSQ- total | 4.84 | 1.74 |
| PSQ-minor | 3.88 | 1.86 |
| PSQ-moderate | 5.30 | 1.79 |
| PCS—Pain catastrophizing | | |
| Total score | 23.46 | 11.48 |
| Rumination | 8.60 | 4.35 |
| Magnification | 4.79 | 2.88 |
| Helplessness | 10.07 | 5.88 |
| TIPI–Personality traits | | |
| Extraversion | 3.47 | 1.46 |
| Agreeableness | 4.98 | 1.05 |
| Neuroticism | 5.91 | 1.08 |
| Conscientiousness | 4.84 | 1.36 |
| Open to experiences | 5.43 | 1.12 |
| Emotional status | | |
| HADS–Anxiety | 6.65 | 4.99 |
| HADS–Depression | 3.80 | 3.24 |
| PSS–Stress | 14.54 | 6.68 |

## 3.2. Correlations between study variables

Some preliminary support for our hypotheses were observed in the Pearson correlation matrix (Table 2). First, PCS and PSQ scores were significantly linked (r = .31; p<0.001). Second, only PSQ scores were directly correlated with acute pain intensity for both the head (r = .22, p = 0.001) and neck (r = .22, p<0.001). Moreover, PCS scores were significantly related to conscientiousness (r = -.27; p<0.001), anxiety (r = .19; p<0.001), depression (r = .29; p<0.001), and stress (r = .37; p<0.001).

## 3.3. Structural Equation Model (SEM) analyses

The first SEM analysis model simultaneously examined the direct and indirect relationships among the latent independent variable pain catastrophizing (rumination, magnification, and helplessness subscales), the latent mediator pain sensitivity (mild and moderate subscales) and acute head and neck pain, adjusting for age and gender (Fig 2).

The data had a good fit to the models according to the fit indices as presented in Fig 2. Pain catastrophizing was significantly associated with pain sensitivity for head and neck pain models (γ = .36 and γ = .37, respectively; p<0.001). In turn, pain sensitivity was significantly related to acute head and neck pain: β = .21, p<0.05 and β = .24, p<0.001, respectively. Both gender and age were not significantly related to head or neck pain.

Alternative models with a direct path between the independent variable pain catastrophizing and head or neck pain found non-significant relationship (p = .686 and p = .193, respectively). Additionally, the difference in chi square and degrees of freedom between the initial models and the alternative models presented were not significant (p = .290 and p = .187, respectively), thus the initial models are the preferred models since they are more parsimonious.

In summary, the first SEM analysis suggests that higher pain catastrophizing is not directly associated with higher acute head and neck pain intensity ratings. However, higher pain catastrophizing was significantly linked to higher pain sensitivity, which in turn was significantly associated with higher acute head and neck pain intensity ratings. Thus, the relationship between high pain catastrophizing and high acute head and neck pain was fully mediated by high pain sensitivity.

The second SEM analysis model included the five personality traits: extraversion, agreeableness, conscientiousness, neuroticism and open to experiences, as independent variables that

**Table 2. Correlations between study variables.**

| Study variables | 1. | 2. | 3. | 4. | 5. | 6. | 7. | .8 | 9. | 10. | 11. | 12. |
|---|---|---|---|---|---|---|---|---|---|---|---|---|
| 1. PCS total | 1 | | | | | | | | | | | |
| 2. PSQ total | .31 * * | 1 | | | | | | | | | | |
| 3. Extraversion | -.06 | .00 | 1 | | | | | | | | | |
| 4. Agreeableness | -.04 | -.00 | -.21 * * | 1 | | | | | | | | |
| 5. Neuroticism | -.00 | -.01 | -.13 | .21 * * | 1 | | | | | | | |
| 6. Conscientiousness | -.27 * * | -.20 * * | -.22 * * | .13 | .09 | 1 | | | | | | |
| 7. Open to experiences | -.09 | -.08 | .06 | .07 | .07 | .12 | 1 | | | | | |
| 8. Anxiety | .19 * * | .13 | .16 * | -.11 | .04 | -.28 * * | -.05 | 1 | | | | |
| 9. Depression | .29 * * | .24 * * | -.05 | -.08 | -.03 | -.24 * * | -.19 * * | .32 * * | 1 | | | |
| 10. Stress | .37 * * | .19 * * | .08 | -.14 * | -.15 * | -.38 * * | -.00 | .34 * * | .39 * * | 1 | | |
| 11. Head Pain Avg | .07 | .22 * * | .04 | .01 | .01 | -.02 | .01 | .05 | .07 | .02 | 1 | |
| 12. Neck Pain Avg | .02 | .22 * * | .02 | -.04 | .02 | .00 | .04 | -.00 | -.04 | .05 | .55 * * | 1 |

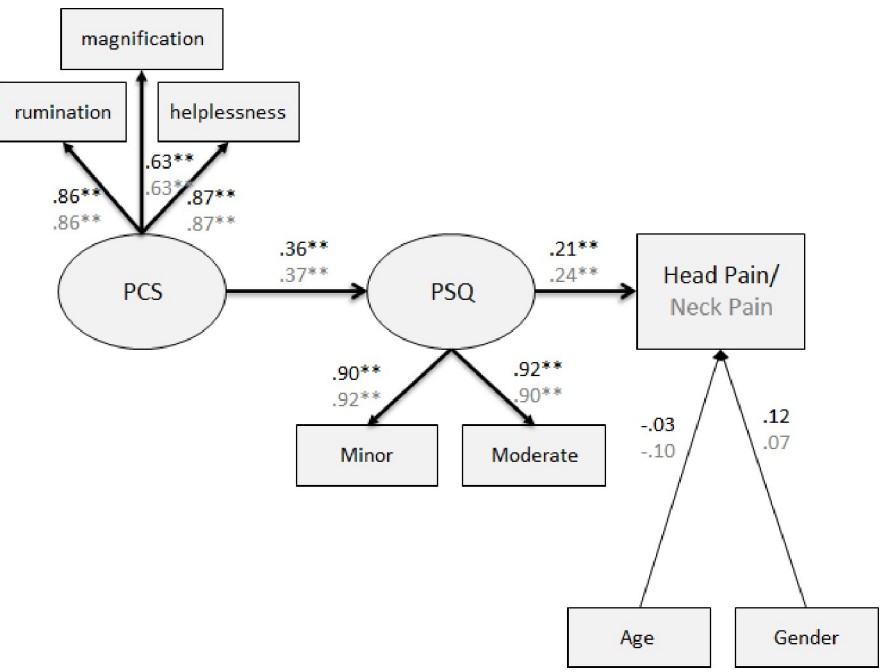

**Note: Fit indices:**
Head pain as outcome: Chi²(df) 27.87(19); CFI .98; TLI .97; RMSEA .04
Neck pain as outcome: Chi²(df) 28.99(19); CFI .98; TLI .97; RMSEA .05
\* p<.05
\*\* p<.001

**Fig 2. SEM analysis of the direct and indirect link between PCS, PSQ and head or neck pain intensity ratings.**
Note: Estimates in black represent the model with head pain as an outcome and estimates in grey represent the model with neck pain as an outcome. *p < .05 **p < .001.

are both directly associated with pain catastrophizing and also correlated with each other (Fig 3). Although the chi square p-values of both models were significant (p = .029 and p = .021, respectively), other fit indices indicated that the data had a good fit to the models as presented in Fig 3.

Only the personality trait of conscientiousness was significantly negatively related to pain catastrophizing γ = -.32 (p<0.001) in both models. As in the initial model that was presented in Fig 2, pain catastrophizing was significantly associated with the pain sensitivity, which in turn was significantly related to both acute head and neck pain intensity ratings.

Alternative models with two new direct paths were examined. The first path was a direct path between conscientiousness and pain sensitivity and the second path was a direct path between conscientiousness and acute head or neck pain intensity ratings. Both new paths were not significant in both head and neck pain models: for the acute head pain model p = .215 for first path and p = .716 for second path; and for the acute neck pain model p = .207 for first path and p = .560 for second path. Additionally, the difference in chi square and degrees of freedom between the models without these additional paths and the alternative models were not significant (p = .206 and p = .169, respectively), thus the models without these additional direct paths are the preferred models.

In summary, the second SEM analysis suggests that personality traits were not directly associated with higher acute head and neck pain intensity ratings. However, the trait of lower conscientiousness was significantly associated with higher pain catastrophizing, which in turn was

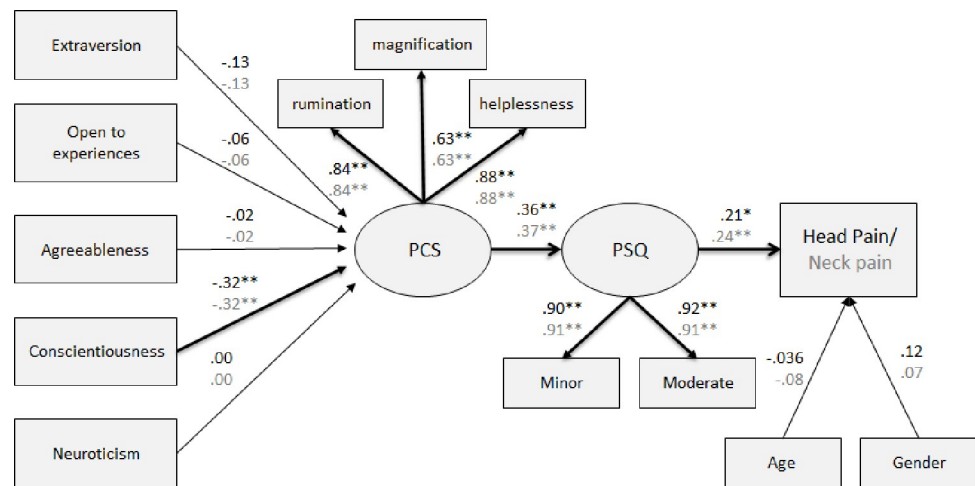

Note: Fit indices:
Head pain as outcome Chi²(df) 75.27(54); CFI .97; TLI .94; RMSEA .04
Neck pain as outcome Chi²(df) 77.28(54); CFI .96; TLI .94; RMSEA .04
\* p<.05
\*\* p<.001

**Fig 3. SEM analysis of the direct and indirect link between personality traits, PCS, PSQ and head or neck pain intensity ratings.** Note: Estimates in black represent the model with head pain as an outcome and estimates in grey represent the model with neck pain as an outcome. *p < .05 **p < .001.

significantly linked to higher pain sensitivity, which in turn was significantly associated with higher acute head and neck pain intensity ratings. Thus, the relationship between low conscientiousness and high acute head and neck pain intensity ratings was fully mediated by high pain catastrophizing and high pain sensitivity.

The third SEM analysis model included an independent latent variable of emotional status, as measured by stress (PSS), anxiety, and depression (HADS), which was directly associated with pain catastrophizing (Fig 4). Fit indices indicated that the data had a good fit to the models. However, the chi square p-values of the model with head pain as dependent variable and the model with neck pain as dependent variable were both significant (p<0.001 and p = .001, respectively).

Emotional status was significantly related to pain catastrophizing in both models (γ = .53, p<0.001), indicating that higher anxiety, depression and/or stress were significantly associated with high pain catastrophizing. In turn, pain catastrophizing was significantly associated with the pain sensitivity (for the acute head pain model β = .38, p<0.001 and for the acute neck pain model β = .39, p<0.001). Finally, higher pain sensitivity was significantly related to higher head and neck pain intensity ratings (β = .20, p<0.001 and β = .24, p<0.001, respectively).

Two new direct paths were entered into alternative models (Fig 5). The first path was a direct path between emotional status and pain sensitivity and the second path was a direct path between emotional status and acute head or neck pain intensity ratings. While the first path was significant for both models (γ = .22, p<0.05 for head pain model and γ = .23, p<0.05 for neck pain model), the second path was not (p = .730 and p = .161, respectively). The difference in chi square and degrees of freedom between the models without these additional paths and the alternative models were significant for the neck but not the head pain intensity model. In the neck pain model, there was a change in the chi square of 6.24 with 2 degrees of freedom,

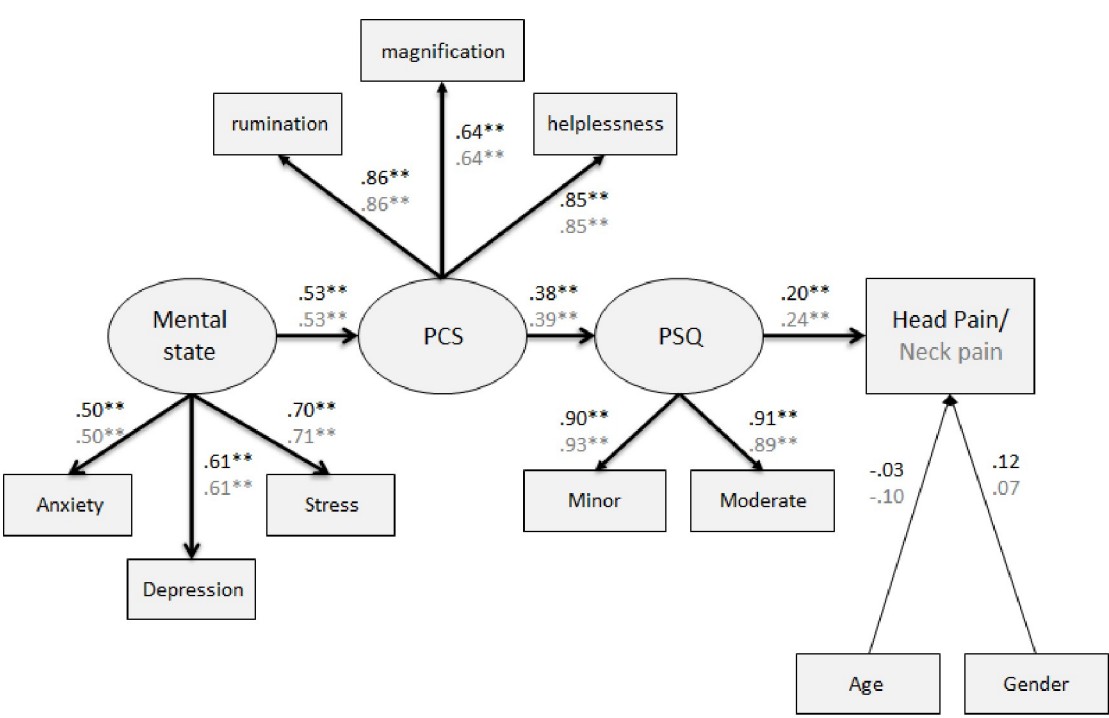

Note: Fit indices:
Head pain as outcome Chi²(df) 78.45(42); CFI .95; TLI .92; RMSEA .06
Neck pain as outcome Chi²(df) 80.29(42); CFI .95; TLI .92; RMSEA .06
\* p<.05
\*\* p<.001

**Fig 4. SEM analysis of the direct and indirect link between emotional status, PCS, PSQ and head or neck pain intensity ratings.** Note: Estimates in black represent the model with head pain as an outcome and estimates in grey represent the model with neck pain as an outcome. *p < .05 **p < .001.

corresponding to p = .044. Thus, the alternative model with these additional direct paths is the preferred one for neck pain intensity model.

In summary, the third SEM analysis suggests just like pain catastrophizing and personality traits, emotional status had no direct association with acute head and neck pain intensity rating. However, unlike conscientiousness, which was not directly linked to pain sensitivity, higher emotional status was significantly associated with higher pain sensitivity. Thus, the relationship between a heightened emotional status and high acute head and neck pain was partially mediated by high levels of pain catastrophizing and fully mediated by high pain sensitivity.

## 4. Discussion

The main findings of this study revealed that solely testing direct links of either situational or dispositional personality characteristics may not be optimal to fully understand acute post-collision pain variability. Rather, pain intensity can be explained by indirect links, through pain sensitivity, with **situational** measures of anxiety, depression and stress, and **dispositional** personality traits, as well as pain catastrophizing.

There is a consensus among scholars in the field of chronic pain that pain catastrophizing is a predictor of pain intensity. However, when we originally explored this factor in our cohort,

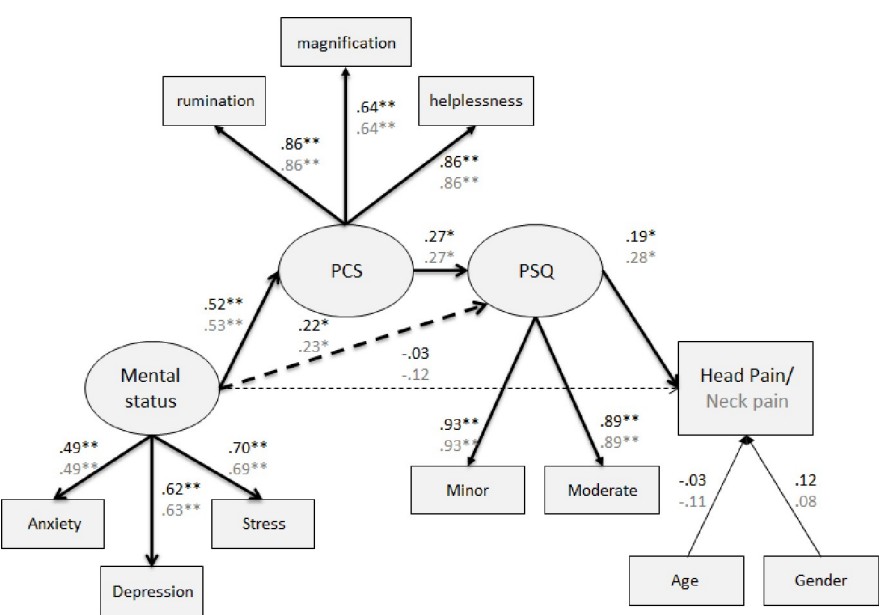

Note: Fit indices:
Head pain as outcome Chi²(df) 74.40(40); CFI .95; TLI .92; RMSEA .06
Neck pain as outcome Chi²(df) 74.05(40); CFI .95; TLI .92; RMSEA .06
* p<.05
** p<.001

**Fig 5. Alternative models of SEM analysis of the direct and indirect link between emotional status, PCS, PSQ and head or neck pain.** Note: Estimates in black represent the model with head pain as an outcome and estimates in grey represent the model with neck pain as an outcome. Strait lines represent original model paths and dashed lines represent additional paths of the alternative model. *p < .05 **p < .001.

we found no correlation between pain catastrophizing and the patients' reported acute pain intensity [17]. To parse these findings with the accumulative available evidence, we then looked to explore whether pain catastrophizing relates to acute pain, via indirect links. Our chosen mediator was pain sensitivity, a relatively new dispositional pain characteristic that has shown correlation with both clinical and experimental pain intensity [34, 43, 52–54]. Unlike other more well-explored pain-related questionnaires, the PSQ is distinct in the task which it requires of the patient. In order to assign a value to the daily-life situations presented within, individual's need to both recall what was, and sometimes imagine what could be if the situation is unfamiliar to them. Thus, within one task an individual summarizes several cognitive representations of emotional and sensory processes which contributes to the pain experience. Indeed, our findings revealed that the hypothesis of mediation was supported, for pain in both the head and neck suggesting that reported pain intensity may be somewhat more of an expression of inherent pain sensitivity, and not only a representation of direct injury.

Since Ruschewyh [43] introduced the PSQ as a tool for pain-related assessment evidence has accumulated to support its contribution to deciphering the nature of 'pain sensitivity'. However, little has been suggested regarding its conceptualization. The theoretical links described in the current findings however help to start elucidating its role, as one which connects different facets of situational and dispositional pain processing. One possible suggestion as to why it can fill a mediating role would be to consider the PSQ as a tool which represents an individual's imagination of pain. Imaging works [55–57] showed that pain imagination is associated with increased activity of brain regions involved in the pain-related neural network.

Thus, it can be assumed that mediating role of PSQ (which reflects deeper cortical representations of the pain matrix) stems from its ability to depict specific routes of pain modulation that shape the manner in which individuals recall, process, and score a pain experience.

Interestingly, it seems that pain catastrophizing may have a dual role in explaining acute pain intensity. Wherein, PCS, when obtained in circumstances of clinical pain, may represent a more situational parameter and therefore is not directly linked with the pain experience, but rather its influence is mediated by the PSQ. At the same time, the PCS also mediated the effect of other dispositional (personality traits) and situational (emotional status) factors on acute pain intensity. While catastrophizing has often been viewed as a personality trait that remains stable in the absence of intervention [e.g., 58, 59], other findings that catastrophizing decreased following pain relief, suggest dynamic, state-like aspects [60]. Thus, the PCS, may comprise both situational and dispositional elements which are affected, among other reasons, by the instructions provided (whether relate to current or on previous pain experience) [61]. Furthermore, Sullivan et al. [58] proposed that the extent to which one engages in catastrophizing might **change over time** as a function of stimulus cues and social responses present in the individual's environment. It is thus possible that certain individuals have a tendency to catastrophize in response to pain but that this tendency is amplified under certain internal conditions (depressed, anxious or stressed). A small number of studies have even begun to examine pain catastrophizing as a mediating factor in the context of acute pain [13, 62], further strengthening this explanation.

In regard to the personality traits themselves, while neuroticism and extraversion have been widely explored in previous pain literature [24, 25], the role of conscientiousness seems to be less investigated. Individuals with high scores of conscientiousness are characterized by high accountability as well as ethical responsibility and trustworthiness [63–65]. Previous work in acute pain settings has explored the role of conscientiousness and health outcomes [66, 67]. Thus, one can assume that individuals who scored higher on conscientiousness also demonstrate "more positive" attitudes when they face pain. A recent study [27] suggest that those with high conscientiousness often engage in more adaptive health management behaviors, which may also reflect increased self-efficacy and control perception. This line of thinking may explain why post-collision participants with low conscientiousness exhibited higher catastrophizing thinking toward pain, because their conscientiousness trait served as a 'buffer' which allowed them to construct a more adaptive cognitive representation towards pain which attenuates the negative meaning and consequence of their pain symptoms. Specifically, and in line with our findings, Suso-Ribera et al., [66] noted that conscientiousness tended to be associated with better health outcomes, including physical functioning and mental well-being, and Conrad and Stricker [67] showed that conscientiousness women demonstrated positive labor experience.

Situational factors, such as emotional status (e.g., depression, anxiety, stress), were previously reported to be linked to both pain catastrophizing [28, 29], and pain sensitivity [33, 34]. As was observed in this cohort, previous work reported that those with state anxiety experienced higher levels of acute pain in the presence of higher catastrophizing. Our work expands on this by proposing that pain catastrophizing partially mediates an individual's overall emotional status, comprised of stress, anxiety and depression, which in turn affects their perception of acute post-injury pain. This perspective of mediating factors is prudent as recent cumulative work has failed to find direct correlations between state anxiety and depression, as reflected by the HADS, and acute post-operative pain [30–32]. Furthermore, it supports very recent findings [23], which are based on the Extended Dynamic Mediation Model that tested an integrative theoretical model of the association between personality traits and trait affect in combination with the dynamic mediation hypothesis [68]. Taken together, it seems that both

PC and pain sensitivity mediate the relations between personality traits and emotional states, and pain experience.

Despite the large clinical cohort and the advanced statistical methods employed several limitations should be noted. First, given that a comprehensive amalgamation of features shape the ability to perceive, cope with, and react to pain, the cross-sectional nature of the current study does may not allow for a full exploration of the multidimension and fluctuation in the magnitude of the assessed variables. Second, the results regarding personality traits should be carefully interpreted due to the use of the TIPI, which is a short-form tool, and has been less explored in the literature as compared with the full Big Five Inventory.

## 5. Conclusion

In conclusion, the wide variability in the manifestation of acute post-traumatic pain can be better understood when addressed not as an isolated concept, but rather as a combination of both dispositional and situational influences. Taking this enriched view, and by use of both direct and indirect pathways, it allows for a more in-depth understanding of factors which may affect the acute to chronic pain transition.

## Supporting information

**S1 Data.**
(SAV)

## Author Contributions

**Conceptualization:** Michal Granot, Yelena Granovsky, David Yarnitsky.

**Data curation:** Einav Srulovici, Pora Kuperman.

**Formal analysis:** Michal Granot, Pora Kuperman.

**Funding acquisition:** Michal Granot, Yelena Granovsky, David Yarnitsky.

**Investigation:** David Yarnitsky, Pora Kuperman.

**Methodology:** David Yarnitsky.

**Project administration:** Michal Granot.

**Supervision:** Michal Granot, Yelena Granovsky.

**Writing – original draft:** Michal Granot, Einav Srulovici, Pora Kuperman.

**Writing – review & editing:** Michal Granot, Einav Srulovici, Yelena Granovsky, David Yarnitsky, Pora Kuperman.

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
