## [Decision Letter · Decision Letter 0]

27 Apr 2021

PONE-D-20-40180

Pain Sensitivity mediates between pain-related personality features and acute mTBI post-collision pain

PLOS ONE

Dear Dr. Granot,

Thank you for submitting your manuscript to PLOS ONE. After careful consideration, we feel that it has merit but does not fully meet PLOS ONE’s publication criteria as it currently stands. Therefore, we invite you to submit a revised version of the manuscript that addresses the points raised during the review process.

We look forward to receiving your revised manuscript.

Kind regards,

Inmaculada Riquelme

Academic Editor

PLOS ONE

Journal Requirements:

PLOS requires an ORCID iD for the corresponding author in Editorial Manager on papers submitted after December 6th, 2016. Please ensure that you have an ORCID iD and that it is validated in Editorial Manager. To do this, go to ‘Update my Information’ (in the upper left-hand corner of the main menu), and click on the Fetch/Validate link next to the ORCID field. This will take you to the ORCID site and allow you to create a new iD or authenticate a pre-existing iD in Editorial Manager. Please see the following video for instructions on linking an ORCID iD to your Editorial Manager account: https://www.youtube.com/watch?v=_xcclfuvtxQ

I have read the journal's policy and the authors of this manuscript have the following competing interests:D. Y. holds equity in BrainsGateLtd. and Theranica Ltd. All other authors report no relevant disclosures or potential conflicts of interest.

Reviewers' comments:

Reviewer's Responses to Questions

**Comments to the Author**

1. Is the manuscript technically sound, and do the data support the conclusions?

Reviewer #1: Partly

Reviewer #2: Partly

2. Has the statistical analysis been performed appropriately and rigorously? 

Reviewer #1: I Don't Know

Reviewer #2: No

3. Have the authors made all data underlying the findings in their manuscript fully available?

Reviewer #1: Yes

Reviewer #2: Yes

4. Is the manuscript presented in an intelligible fashion and written in standard English?

Reviewer #1: Yes

Reviewer #2: No

5. Review Comments to the Author

Reviewer #1: The study focuses on acute pain in non-post-operative patients following mild traumatic brain injury and attempts to untangle the effects of pain-related psychological measures and personality traits on the acute pain perception. It is an interesting topic, and the sample is quite rich. Nevertheless, I have several concerns regarding the current version of the manuscript.

Introduction:

I would suggest expending introduction to better articulate the aim of the study. Namely, it would be useful to expend on previous findings on the relationship between pain perception and personality traits (currently lines 77-87 just reads that some relatively stable personality characteristics are related to pain). This will enable you to make better hypothesis – i.e. „positive“ and „negative“ personality traits is vague and not very accurate depiction of what you are testing here (my overall impression is that the authors do not come from the field of personality which resulted in errors in this aspect of the paper – see below). Please do not label personality traits as negative/positive – each trait is a continuum and high/low scores are not good or bad (depending on the context, both high and low scores can be beneficial for the person, different societies value different expressions of different traits, often „mid-range“ scores are the most adaptive, and extremes (on either low/high) can be dysfunctional.

Although unusual, I like how the hypothesis are embedded in the introduction. However, I am missing the objective of the paper – Why are we contrasting these models? The first aim is clear to me, but the second aim („to investigate how the first model is affected by personality) and the third aim (how the first model is affected by emotional states) are something I am struggling to understand: What is the rationale for this? Why is this something we should be looking into? (I can use my imagination but it is always better to have it explicitly stated in the paper)

Methods

I think that the headings should be differently ordered, Eg. I don’t see how „Study population“ (second order heading) comprises Participants & Study design. I would recommend 2.1. Study design 2.2. Participants 2.3. Measures 2.4. Statistical analysis. Under 2.3. you can have the „primary outcome“ and „predictor measures“, but please do not label TIPI as „pain-related personality questionnaire“; same goes for HADS and PSS.

As this study is a part of a large ongoing data collection, and you have already published some data in other papers, please explain how data presented in this paper differ form the datasets that have already been published. This is something one should pay a lot of attention to in order to avoid double publishing and/or salami-slicing (this is one of the critical issues for me regarding this paper)

On the similar note, you need to justify the sample size – please include the stopping rule (i.e. since the study is ongoing, how did you decide to take this set of participants – why October 2018?, did the study stop for some reason, or was there something else. 200 is a very round number – did you stop at 200 participants?); also power calculation would be useful here so I would strongly suggest moving it from statistical analysis to sample section, and adding the sample size here as well (it is more reader-friendly to have sample description size and power at the same place rather than pages apart)

It is not clear to me how the primary outcome was calculated. Line 152-153 authors state the mean rating for both (read: head and neck), were considered primary outcome measures. What was averaged? The lines 146-147 state: mean pain in the neck, mean pain in the head, maximum pain in the neck, maximum pain in the head. Was it the mean between “mean” and “maximum”, or was just the mean measure used? Why was this measure selected as primary outcome? If other measures have been collected but not used as outcomes, I think it would be good to add the rationale behind that decision or add the analysis on maximal pain to the results in table 2 for example.

Please avoid using “pain-related personality” - Pain Sensitivity Questionnaire is not assessing personality, and TIPI is not assessing pain.

Pain Catastrophizing Scale – please add the information on which scores have been used, the total or subscales? “Patients were not directed to focus on any particular pain sensation”- please provide the rationale behind this decision.

Five Feature Model (FFM) is actually Five Factor Model – please correct this.

In relation to TIPI, I have to ask, just to be sure – were the reverse coded items recoded before averaging items for each factor?

The sentence “…. with high scores endorsing a stronger affirmation of the personality dimension and each dimension is corelated with the other” is not correct. One cannot say “stronger affirmation of dimension” – high score on extraversions means that the person is extraverted, and low means that he/she is more reserved or quiet, so the “baseline” is the middle, not the lower score. The main thing about these five dimensions (and the whole idea behind FFM) is that they are relatively independent – they can show some correlations, but these correlations are low, but they are meant to be mainly independent.

Please consult the paper on TIPI development Gosling, S. D., Rentfrow, P. J., & Swann, W. B., Jr. (2003). A Very Brief Measure of the Big Five Personality Domains. Journal of Research in Personality, 37, 504-528, and also the main works on Big Five / Five Factor Model by Goldberg and Costa & McCrae.

“Anxiety often precedes depression in response to stressors and is often poorly identified by clinicians” – this is a bit of a strong claim, please either support it by empirical evidence or simply omit it.

Please add the reliability measures for all instruments (e.g. internal consistency either at this sample or from the previous studies), this is important for the subsequent analysis.

the “chi-square distribution” should be “chi-square test”

replace “ the χ2 estimated value was low and the p-value was greater than .05” with “non-significant chi-square test”

SEM stands for Structural Equation Modeling – so please adjust the wording in the sentences e.g. replace “SEM analyses were tested” with “models were tested using SEM”.

Results

When language is important for conducting the study, it should be included in the inclusion/exclusion criteria, because when one needs to fill in the questionnaire, he/she did not dop out due to the language barrier, but he/she was not supposed to be included in the study at all.

It is important to comment on the fact that the study is underpowered for testing the second model.

I find SDs for TIPI seem a bit small – could you please check the data once more and make sure that they are correct.

SD should be reported with 2 or even 3 decimal places.

Why correlations with only some of the questionnaires and not all are presented?

What do you mean by “which is in line with the previous findings for this cohort”? Is it the same sample, different sample or partially overlapping sample?

Please avoid using “pain-related personality factors” throughout the manuscript.

“correlator” should be replaced with “corelate” line 275

Please report on chi-squares and respective degrees of freedom for models.

It is important not to alter between Neuroticism and Emotional stability as they are the “opposites”. From the paper it is not clear to me whether higher scores on this dimension reflect stability or neuroticism. Please adjust the text and provide this information explicitly.

Please label personality traits as they are commonly labeled or how they have been labeled in TIPI eg. Openness is not Open to changes

Looking at the model 2 that was tested - I am wondering why all five traits were included in the model (and this is why I suggested to explain the introduction) – it is reasonable to assume that N would be related, also A makes some sense, early work on E would suggest so as well, but I a cannot grasp the expectation regrading O for example. This is why it is important to convince the reader that what you are testing has some grounds in the previous literature, or if that is not the case - explicitly state which hypothesis you are testing and what is the rationale behind it.

I am missing zero order correlations between personality, depression, stress, and pain measures, because they are important for understanding the results.

It is not clear what is the operational definition of higher/lower emotional status, please explain.

Discussion

For a number of reasons, I believe it important not to label depression as “situational measure”, same goes for anxiety, even stress is not something that can be easily labeled as situational measure. Depression is deeply rooted in our neurobiology; it is not situationally driven nor measured as situational variable. It is also very important not to make situation-disposition distinction between for eg. depression-neuroticism (just do the correlation analysis between the two on the data set you have, and you’ll see why such distinction cannot be made)

I don’t see the relevance of discussing imagination and pain matrix in the contest of current study. I am not sure if I am missing something or if this part should be excluded altogether.

The relationship between C and health outcomes is usually attributed to the higher adherence to the recommendations and more orderly lifestyle (but I am not sure that such explanation would fit here). The role of C is not discussed adequately – I believe it is important to provide some explanation on what this finding could mean and why it was obtained.

It is unclear how did you derived the following conclusion “Taken together, it seems that personality states, experienced situation characteristics, and state affect mediate the relations between personality traits and trait affect.” Also, please try to be more consistent with the terminology – what are “personality states” “experienced situation characteristics” “state affect” ?

While I would personally agree that causal inference should be based on experiments i.e. manipulation of the conditions, the analysis you are preforming here is the closes you can get to testing causal effects when all variables are registered rather than manipulated. See for example: https://ftp.cs.ucla.edu/pub/stat_ser/r370.pdf and adjust that part of discussion on limitations (I would not attribute the limitation to the cross-sectional design, I see the sentence that follows it but these two sentences are in collision).

It is difficult to draw conclusions on the “situational influences” as none of the variables in the study was situational.

Overall, I believe this is an interested topic, and you have a good data set. From the prospect of publishing the critical issue is how you extract different papers for the same data set – so it is important to be clear and open here so that the same data is not reported several times. On the same note – I find it essential provide more detail regarding the sample selection and the inclusion criteria (I am just confused – that the study is ongoing and that you decided not to use any data collected in the past two years). On the content side, I believe it is essential to correct how the data on personality, depression, and stress are presented. This would require diving into at least most prominent papers on the personality models, depression measurement and interpretation of those measures and doing some literature research on how all these psychological variables relate to each other. If you see the merit of revising your paper in that direction, I will be happy to review again after resubmission.

Reviewer #2: The paper “Pain Sensitivity mediates between pain-related personality features and acute mTBI post-collision pain” presents interesting findings on the mediation effects of pain sensitivity on the relationship between specific personality features and acute mTBI post-collision pain.

The study has both practical and scientific relevance. However, several issues deserve paying attention to before the paper can be accepted for publication.

Overall, the report needs a lot of polishing and structuring. It contains a lot of valuable information but it is not presented in a user-friendly manner. It requires a lot of repeated reading to grasp the presented information.

Obviously, there is an error in the labels of the figures, as all figures are labeled as Figure 1 – this has to be corrected. Also, please check carefully the text in which figures are mentioned.

In Table 2 authors give correlations between a set of measures, but not for all measures. Personality traits are completely excluded from the Table as if they are not relevant. If so, why personality traits were explored at all in subsequent analyses? Please add correlations between personality traits and pain measures as discuss them. Additionally, tables could be improved – for example, non-significant p-values do not have to be displayed.

The introduction is, in my opinion, rather poor in displaying available evidence and discussing why authors decided to include both traits and states and how they relate to pain sensitivity. I recommend careful revision of the introduction. Also, I would recommend authors to use more precise terminology (e.g., terms “positive” and “negative” personality traits should be more precise, to what traits authors refer to).

Hypothesis 2 – authors introduce terms “positive” and “negative” personality traits without specifying to which traits they refer. Thus, judging the quality of the hypothesis is very difficult. Additionally, as part of the text elaborating on Hypothesis 2, the authors introduce negative affective states, which are not part of the hypothesis. Since affective states are related to personality traits, it has to be justified why personality traits and affective states were analyzed separately. Why did the authors decide to pursue that kind of analytic strategy?

Hypothesis 3 is completely unexplained – it is not clear why the Hypothesis is formulated that way, and what is meant under the term “A heightened post-collision emotional status”.

Did the authors try to explore both personality traits and states in one model)?

I would recommend authors to structure the presentation of the results – fit indices of tested models and comparison of models can be displayed in a table.

In all tested models, age and gender are postulated as relevant factors, but the introduction is not saying much about the relevance of sociodemographic variables on the criterion variable. Please revise the text and explain why we should focus on age and gender differences.

I applaud the authors for making their dataset available, but I recommend them to add labels of the variables, and values for each variable. Also, adding a CSV file would increase the visibility and transparency of the dataset, as the .sav file requires licensed software.

6. PLOS authors have the option to publish the peer review history of their article (what does this mean?). If published, this will include your full peer review and any attached files.

Reviewer #1: **Yes: **Jovana Bjekic

Reviewer #2: **Yes: **Ljiljana B. Lazarevic

---

## [Author Response · Author response to Decision Letter 0]

13 Aug 2021

Dear Jovana Bjekic and Ljiljana B. Lazarevic,

We thank you for taking the time to review our manuscript entitled “Pain Sensitivity mediates between pain-related personality features and acute mTBI post-collision pain”. We have taken painstaking measures to address your insightful comments, and hope that the manuscript is now clearer and more understandable. Please find our responses below.

Sincerely yours,

The authors 

PONE-D-20-40180

Pain Sensitivity mediates between pain-related personality features and acute mTBI post-collision pain

PLOS ONE

Reviewer #1: 

The study focuses on acute pain in non-post-operative patients following mild traumatic brain injury and attempts to untangle the effects of pain-related psychological measures and personality traits on the acute pain perception. It is an interesting topic, and the sample is quite rich. Nevertheless, I have several concerns regarding the current version of the manuscript.

Introduction:

I would suggest expending introduction to better articulate the aim of the study. Namely, it would be useful to expend on previous findings on the relationship between pain perception and personality traits (currently lines 77-87 just reads that some relatively stable personality characteristics are related to pain). This will enable you to make better hypothesis – i.e. „positive“ and „negative“ personality traits is vague and not very accurate depiction of what you are testing here (my overall impression is that the authors do not come from the field of personality which resulted in errors in this aspect of the paper – see below). Please do not label personality traits as negative/positive – each trait is a continuum and high/low scores are not good or bad (depending on the context, both high and low scores can be beneficial for the person, different societies value different expressions of different traits, often „mid-range“ scores are the most adaptive, and extremes (on either low/high) can be dysfunctional.

Our response: the reviewer's comment is very important, and it guided us in the revision of the manuscript in general and the introduction section in particular. We re-wrote the entire introduction section according to this comment. We hope that the new version addresses the expectations of the reviewers and allowed us to improve our manuscript. Please see Pages 4-10. 

Although unusual, I like how the hypothesis are embedded in the introduction. However, I am missing the objective of the paper – Why are we contrasting these models? The first aim is clear to me, but the second aim („to investigate how the first model is affected by personality) and the third aim (how the first model is affected by emotional states) are something I am struggling to understand: What is the rationale for this? Why is this something we should be looking into? (I can use my imagination but it is always better to have it explicitly stated in the paper)

Our response: Thank you for this comment. Accordingly, in the revised version, we explained that two theoretical models guided this study. Thus, the introduction section was expended, clarified, and current publications were added to strengthen our claims. Please see pages 7 and 8. 

Methods

I think that the headings should be differently ordered, Eg. I don’t see how „Study population“ (second order heading) comprises Participants & Study design. I would recommend 2.1. Study design 2.2. Participants 2.3. Measures 2.4. Statistical analysis. Under 2.3. you can have the „primary outcome“ and „predictor measures“, but please do not label TIPI as „pain-related personality questionnaire“; same goes for HADS and PSS.

Our response: The order of the headings was changed as recommended. We believe that the new order will allow the readers better understanding about the information provided in the Methods section. In addition, we reframed the variable sub-titles such that each variable was defied according of its role in the model: dependent, independent and mediator. Please see pages 11-18. 

As this study is a part of a large ongoing data collection, and you have already published some data in other papers, please explain how data presented in this paper differ form the datasets that have already been published. This is something one should pay a lot of attention to in order to avoid double publishing and/or salami-slicing (this is one of the critical issues for me regarding this paper).

Our response: Although 3 papers were published during the past two years based on data obtained in this research project, this manuscript is unquestionably not a duplication of other reports. First and foremost this is the only analysis of the data, to date, which includes both patients classified as post-whiplash and those classified as post-mTBI, all other papers only focused on a subsection of patients who were considered mTBI post-collision. In addition the first paper concerned itself with a general comparison of healthy controls with mTBI post-collision patients, the second looked to the PSQ as an additive tool for understanding acute pain, and the third, recently published only looked at 6-month follow-up data from a very small subsection of the cohort. 

On the similar note, you need to justify the sample size – please include the stopping rule (i.e. since the study is ongoing, how did you decide to take this set of participants – why October 2018?, did the study stop for some reason, or was there something else. 200 is a very round number – did you stop at 200 participants?); also power calculation would be useful here so I would strongly suggest moving it from statistical analysis to sample section, and adding the sample size here as well (it is more reader-friendly to have sample description size and power at the same place rather than pages apart). 

Our response: We thank the reviewer for raising such important issue. Furthermore, this comment encouraged us to add all patients that enrolled in this study project that was completed in December 2019. Since 2020 we were focused only on the follow-up sessions, which occur at 6 and 12-months post-accident. Accordingly, the revised manuscript is comprised of 239 mTBI patients. This allowed us to re-analyze the 3 models. The information about the time frame in which data was collected has been revised in the Method Section, see Page 11 under Study Design. Indeed, the new analyses further support our initial findings and the revised version showed a better model fit in all analyses. Please see at the end of the statistical analyses section the specific information about the sample size required for each model as appropriated in SEM ( Page 19), as this is where we found the information traditionally to be located. 

It is not clear to me how the primary outcome was calculated. Line 152-153 authors state the mean rating for both (read: head and neck), were considered primary outcome measures. What was averaged? The lines 146-147 state: mean pain in the neck, mean pain in the head, maximum pain in the neck, maximum pain in the head. Was it the mean between “mean” and “maximum”, or was just the mean measure used? Why was this measure selected as primary outcome? If other measures have been collected but not used as outcomes, I think it would be good to add the rationale behind that decision or add the analysis on maximal pain to the results in table 2 for example.

Our response: We agree that the description of the outcome measure was not clear. Therefore, we rephrased this point as suggested and state in the revised version that the outcome measures represent the mean pain intensity ratings as assessed by VAS. Please see Page 13. 

Please avoid using “pain-related personality” - Pain Sensitivity Questionnaire is not assessing personality, and TIPI is not assessing pain.

Our response: According to this comment, we corrected this phrase along the revised manuscript in the relevant places. 

Pain Catastrophizing Scale – please add the information on which scores have been used, the total or subscales? 

Our response: The 3 models included the three sub-scales of PCS and not the total PCS scores as appropriated in SEM. This is now noted on Page 16. 

“Patients were not directed to focus on any particular pain sensation”- please provide the rationale behind this decision.

Our response: Since we were interested in ascertaining patient’s general potential catastrophizing view of painful events, patients were instructed to refer to any commonly experienced previous painful event when they completed the PCS. This approach is commonly used in the assessment of pain catastrophizing as suggested by Sullivan et al. (1995). We have better explained how patients were instructed and state that this is according to Sullivan's direction. Please see Page 16. 

Five Feature Model (FFM) is actually Five Factor Model – please correct this.

In relation to TIPI, I have to ask, just to be sure – were the reverse coded items recoded before averaging items for each factor?

Our response: This typo was corrected. As for the second question, the reverse coded items were recoded as required.

The sentence “…. with high scores endorsing a stronger affirmation of the personality dimension and each dimension is corelated with the other” is not correct. One cannot say “stronger affirmation of dimension” – high score on extraversions means that the person is extraverted, and low means that he/she is more reserved or quiet, so the “baseline” is the middle, not the lower score. The main thing about these five dimensions (and the whole idea behind FFM) is that they are relatively independent – they can show some correlations, but these correlations are low, but they are meant to be mainly independent. Please consult the paper on TIPI development Gosling, S. D., Rentfrow, P. J., & Swann, W. B., Jr. (2003). A Very Brief Measure of the Big Five Personality Domains. Journal of Research in Personality, 37, 504-528, and also the main works on Big Five / Five Factor Model by Goldberg and Costa & McCrae.

Our response: Based on this comment, the revised manuscript emphasizes that each dimension is independent. Additionally, we added information regarding that the middle point of each continuum for each trait is considered as the baseline score of this factor. Please see page 14. 

“Anxiety often precedes depression in response to stressors and is often poorly identified by clinicians” – this is a bit of a strong claim, please either support it by empirical evidence or simply omit it.

Our response: According to this comment we have now omitted this sentence

Please add the reliability measures for all instruments (e.g. internal consistency either at this sample or from the previous studies), this is important for the subsequent analysis.

Our response: This information was added to the Methods sections for each of the Instruments. 

The “chi-square distribution” should be “chi-square test”

replace “ the χ2 estimated value was low and the p-value was greater than .05” with “non-significant chi-square test”

Our response: This has been corrected as suggested by the reviewer

SEM stands for Structural Equation Modeling – so please adjust the wording in the sentences e.g. replace “SEM analyses were tested” with “models were tested using SEM”.

Our response: This has been corrected as suggested by the reviewer

Results

When language is important for conducting the study, it should be included in the inclusion/exclusion criteria, because when one needs to fill in the questionnaire, he/she did not dropout due to the language barrier, but he/she was not supposed to be included in the study at all.

Our response: Thank you for this comment, which is very important. Accordingly, we added information about language to the exclusion criteria on Page 13, in that participants who could not communicate in Hebrew were not enrolled into the study.

It is important to comment on the fact that the study is underpowered for testing the second model.

Our response: According to your previous comments about the sample, we now re-analyzed the full data with all the participants in the entire study. Thus, the revised version fully addressed the power requirement.

I find SDs for TIPI seem a bit small – could you please check the data once more and make sure that they are correct.

Our response: We double checked the raw data from the TIPI and found no typo in it. 

SD should be reported with 2 or even 3 decimal places.

Our response: Done. We have added 2 decimal places as suggested in the text and Table 1. Additionally, correlations and p-values are now presented with 3 decimal places.

Why correlations with only some of the questionnaires and not all are presented?

Our response: The revised manuscript now contains the full correlation matrix as suggested, see Table 2.

What do you mean by “which is in line with the previous findings for this cohort”? Is it the same sample, different sample or partially overlapping sample?

Our response: As mentioned previously, in relate to your concern about the potential overlapping, although reports based on part of this cohort were published previously, the current manuscript is focused on a different study question, analyzed in a different statistical approach, in a larger sample and was based on measures that were not published before (for example TIPI). 

Please avoid using “pain-related personality factors” throughout the manuscript.

“correlator” should be replaced with “corelate” line 275

Our response: This has been corrected. 

Please report on chi-squares and respective degrees of freedom for models.

Our response: This has been done. 

It is important not to alter between Neuroticism and Emotional stability as they are the “opposites”. From the paper it is not clear to me whether higher scores on this dimension reflect stability or neuroticism. Please adjust the text and provide this information explicitly.

Please label personality traits as they are commonly labeled or how they have been labeled in TIPI eg. Openness is not Open to changes

Our response: Thank you for this astute comment. We have made changes to the wording concerning the TIPI variables, see for example Table 1. 

Looking at the model 2 that was tested - I am wondering why all five traits were included in the model (and this is why I suggested to explain the introduction) – it is reasonable to assume that N would be related, also A makes some sense, early work on E would suggest so as well, but I a cannot grasp the expectation regrading O for example. This is why it is important to convince the reader that what you are testing has some grounds in the previous literature, or if that is not the case - explicitly state which hypothesis you are testing and what is the rationale behind it.

Our response: In line with this comment and previous ones, we revised the background section such that the new version better elaborates the theoretical and empirical rational for our hypotheses. 

I am missing zero order correlations between personality, depression, stress, and pain measures, because they are important for understanding the results.

Our response: The revised manuscript now includes a full –correlation table for all assessed variables.

It is not clear what is the operational definition of higher/lower emotional status please explain.

Our response: We have clarified that emotional status is a latent variable that was attained by PSS and HADS and that high scores in these questionnaires represents higher emotional distress. Please see Page 14. 

Discussion

For a number of reasons, I believe it important not to label depression as “situational measure”, same goes for anxiety, even stress is not something that can be easily labeled as situational measure. Depression is deeply rooted in our neurobiology; it is not situationally driven nor measured as situational variable. 

Our response: This comment is very important and thanks to the reviewer’s comment we addressed the perception of this term even in the introduction section. In line with your example about depression, we clarified that dispositional and situational measures are indeed two distinctive features but as seen in depression, situational measures might be manifested in more flexible manner then dispositional measures, such that under particular circumstances, the expression of situational features may be altered after an exposure to a demanding situation in either short- or long-term manner. 

It is also very important not to make situation-disposition distinction between for eg. depression-neuroticism (just do the correlation analysis between the two on the data set you have, and you’ll see why such distinction cannot be made)

Our response: We absolutely agree with that comment. Accordingly, this was omitted from the manuscript.

I don’t see the relevance of discussing imagination and pain matrix in the contest of current study. I am not sure if I am missing something or if this part should be excluded altogether.

Our response: According to this comment, we revised the paragraph that now better explains the mediating role of PSQ, which can be found on Page 29.

The relationship between C and health outcomes is usually attributed to the higher adherence to the recommendations and more orderly lifestyle (but I am not sure that such explanation would fit here). The role of C is not discussed adequately – I believe it is important to provide some explanation on what this finding could mean and why it was obtained.

Our response: Indeed, the role of C in pain processing and experience was less explored and reported in the current literature. We assume that individuals who scored higher on C demonstrate also “more positive” attitudes as well as practices when pacing challenging health condition including pain. This assumption is based on a recent study (Day et al., 2021) who stated that :

“Prior research has also shown that more conscientious individuals have more positive health perceptions and visits to the doctor (Jerram & Coleman, 1999), as well as more positive attitudes toward orthodontic treatment which was related to treatment success (Singh et al., 2017). Hence, highly conscientious individuals may tend to evidence

more adaptive health management behaviors, which may also be reflected by higher levels of self-efficacy and perceived control.”

In line with this, the negative association between C and PCS which leads to enhanced pain can be explained by the “buffered” role of this particular trait which allow individuals to construct more adaptive cognitive representation toward pain which attenuate its negative meaning and consequence. We better elaborate this notion in the revised discussion section, please see Page 30. 

It is unclear how did you derived the following conclusion “Taken together, it seems that personality states, experienced situation characteristics, and state affect mediate the relations between personality traits and trait affect.” Also, please try to be more consistent with the terminology – what are “personality states” “experienced situation characteristics” “state affect” ? Our response: This sentence has been changed to read: “Taken together, it seems that both PC and pain sensitivity mediate the relations between personality traits and emotional states, and pain experience.”

While I would personally agree that causal inference should be based on experiments i.e. manipulation of the conditions, the analysis you are preforming here is the closes you can get to testing causal effects when all variables are registered rather than manipulated. See for example: https://ftp.cs.ucla.edu/pub/stat_ser/r370.pdf and adjust that part of discussion on limitations (I would not attribute the limitation to the cross-sectional design, I see the sentence that follows it but these two sentences are in collision).

Our response: Thank you for this valuable comment that allowed us to better address, in the revised manuscript, the limitations associated with our study design. We have now clarified that the cross sectional nature of out study limited the ability to depict multidimension and fluctuation in the magnitude of the assessed variables. Please see Page 31. 

It is difficult to draw conclusions on the “situational influences” as none of the variables in the study was situational.

Our response: We believe that the remarkable changes that were performed along the revised manuscript allow the readers to understand the term “situational influences” in a more clarity manner.

Overall, I believe this is an interested topic, and you have a good data set. From the prospect of publishing the critical issue is how you extract different papers for the same data set – so it is important to be clear and open here so that the same data is not reported several times. On the same note – I find it essential provide more detail regarding the sample selection and the inclusion criteria (I am just confused – that the study is ongoing and that you decided not to use any data collected in the past two years). 

Our response: We hope that the changes that were made according to this comment and previous ones illuminate this issue. 

On the content side, I believe it is essential to correct how the data on personality, depression, and stress are presented. This would require diving into at least most prominent papers on the personality models, depression measurement and interpretation of those measures and doing some literature research on how all these psychological variables relate to each other. If you see the merit of revising your paper in that direction, I will be happy to review again after resubmission.

Our response: Thank you for your insightful suggestions along with your thorough comments, the revised manuscript expresses the complexity associated with the assessed psychological variables. We hope that the additional literature we added as well as the clarifications will address your expectations. Please note the extensive addition of new references. 

 

Reviewer #2: The paper “Pain Sensitivity mediates between pain-related personality features and acute mTBI post-collision pain” presents interesting findings on the mediation effects of pain sensitivity on the relationship between specific personality features and acute mTBI post-collision pain.

The study has both practical and scientific relevance. However, several issues deserve paying attention to before the paper can be accepted for publication.

Overall, the report needs a lot of polishing and structuring. It contains a lot of valuable information but it is not presented in a user-friendly manner. It requires a lot of repeated reading to grasp the presented information.

Our response: Thank you for this valuable comment. We have revised the entire manuscript to now better emphasize the theoretical framework for this research as well as rephrased the definitions of the assessed concepts . We hope that the new version is clearer and user-friendly.

Obviously, there is an error in the labels of the figures, as all figures are labeled as Figure 1 – this has to be corrected. Also, please check carefully the text in which figures are mentioned. 

Our response: This has been corrected 

In Table 2 authors give correlations between a set of measures, but not for all measures. Personality traits are completely excluded from the Table as if they are not relevant. If so, why personality traits were explored at all in subsequent analyses? Please add correlations between personality traits and pain measures as discuss them. Additionally, tables could be improved – for example, non-significant p-values do not have to be displayed.

Our response: In line with this comment and reviewer’s #1 comment we have added all study variables to the correlation table.

The introduction is, in my opinion, rather poor in displaying available evidence and discussing why authors decided to include both traits and states and how they relate to pain sensitivity. I recommend careful revision of the introduction. Also, I would recommend authors to use more precise terminology (e.g., terms “positive” and “negative” personality traits should be more precise, to what traits authors refer to).

We agree that the previous introduction section necessitated remarkable revision. Thus, we re-wrote the introduction and emphasized the conceptualization of the main assessed variables as well as the rationale for the study hypotheses. 

Hypothesis 2 – authors introduce terms “positive” and “negative” personality traits without specifying to which traits they refer. Thus, judging the quality of the hypothesis is very difficult. Additionally, as part of the text elaborating on Hypothesis 2, the authors introduce negative affective states, which are not part of the hypothesis. 

Our response: This has been corrected as suggested.

Since affective states are related to personality traits, it has to be justified why personality traits and affective states were analyzed separately. Why did the authors decide to pursue that kind of analytic strategy?

Our response: We agree with the reviewer about this concern, and we believe that this issue can be addressed from several perspectives, each represent different theoretical approach. The debate among scholars regarding this issue, is also present in reviewer #1 comment that states the minimal to no link between traits and states. Additionally, due to power consideration, we avoid additional analysis that comprised both elements. Nevertheless, we revised the introduction section in the new manuscript such that the rational to pursue that kind of analytic strategy was clarified. 

Hypothesis 3 is completely unexplained – it is not clear why the Hypothesis is formulated that way, and what is meant under the term “A heightened post-collision emotional status”.

Did the authors try to explore both personality traits and states in one model)?

Our response: We apology that the terms personality traits and emotional states were presented in confusing manner. Given that each represent different facet of pain perception and modulation, we aimed to explore each demotion as well as to investigate their composition using the mediation analyses models. We believe that the new version addresses your valuable comment.

I would recommend authors to structure the presentation of the results – fit indices of tested models and comparison of models can be displayed in a table.

Our response: In line with this comment, we omitted information about the fit indices from the text and added it to the figures, which we hope clarifies the information. 

In all tested models, age and gender are postulated as relevant factors, but the introduction is not saying much about the relevance of sociodemographic variables on the criterion variable. Please revise the text and explain why we should focus on age and gender differences.

Our response: According to this comment we have added an explanation of using those variables as control variables in the methods section, Please see Page 17.

I applaud the authors for making their dataset available, but I recommend them to add labels of the variables, and values for each variable. Also, adding a CSV file would increase the visibility and transparency of the dataset, as the .sav file requires licensed software.

Our response: Thank you for your positive comment. Accordingly we have arranged the variables’ labels in the dataset.

---

## [Decision Letter · Decision Letter 1]

15 Sep 2021

PONE-D-20-40180R1Dispositional and situational personal features and acute post-collision head and neck pain: Double mediation of pain catastrophizing and pain sensitivityPLOS ONE

Dear Dr. Granot,

Thank you for submitting your manuscript to PLOS ONE. After careful consideration, we feel that it has merit but does not fully meet PLOS ONE’s publication criteria as it currently stands. Therefore, we invite you to submit a revised version of the manuscript that addresses the points raised during the review process.

Althought all the reviewers' remarks have been addressed, Reviewer 2 has detected small inaccuracies in the text that must be corrected prior to the publication.

We look forward to receiving your revised manuscript.

Kind regards,

Inmaculada Riquelme

Academic Editor

PLOS ONE

Journal Requirements:

Reviewers' comments:

Reviewer's Responses to Questions

**Comments to the Author**

1. If the authors have adequately addressed your comments raised in a previous round of review and you feel that this manuscript is now acceptable for publication, you may indicate that here to bypass the “Comments to the Author” section, enter your conflict of interest statement in the “Confidential to Editor” section, and submit your "Accept" recommendation.

Reviewer #1: All comments have been addressed

Reviewer #2: All comments have been addressed

2. Is the manuscript technically sound, and do the data support the conclusions?

Reviewer #1: Yes

Reviewer #2: Yes

3. Has the statistical analysis been performed appropriately and rigorously? 

Reviewer #1: Yes

Reviewer #2: Yes

4. Have the authors made all data underlying the findings in their manuscript fully available?

Reviewer #1: Yes

Reviewer #2: Yes

5. Is the manuscript presented in an intelligible fashion and written in standard English?

Reviewer #1: Yes

Reviewer #2: Yes

6. Review Comments to the Author

Reviewer #1: I believe authors have addressed all the major concerns in a reasonable manner; I like that they reanalyzed the data and added new participants. The intro and discussion are much better written now.

Reviewer #2: Thank you for making changes in the manuscript. I identified some minor issues that should be fixed:

1. Check reference mentioned in line 58 – reference number 6 is missing.

2. Openness to changes should be openness to experiences – please revise it in whole manuscript

3. Encore – wrong spelling, it should anchors

4. Correlations should be presented with 2 decimals.

7. PLOS authors have the option to publish the peer review history of their article (what does this mean?). If published, this will include your full peer review and any attached files.

Reviewer #1: **Yes: **Jovana Bjekic

Reviewer #2: **Yes: **Ljiljana B. Lazarevic, University of Belgrade, Serbia

---

## [Author Response · Author response to Decision Letter 1]

24 Sep 2021

PONE-D-20-40180R1

Dispositional and situational personal features and acute post-collision head and neck pain: Double mediation of pain catastrophizing and pain sensitivity

PLOS ONE

Dear Jovana Bjekic and Ljiljana B. Lazarevic,

We would like to thank you for taking the time to review our manuscript entitled “Dispositional and situational personal features and acute post-collision head and neck pain: Double mediation of pain catastrophizing and pain sensitivity”. We have made the minor changes that were requested and hope that the editor and reviewers will be satisfied with this version of the manuscript. Please find our responses below.

Sincerely yours,

The authors 

 

Editorial comments:

Our response: Thank you for pointing this out. We have updated many recent citations and removed some of the older ones during the previous revision. This process resulted in an error in the reference list. Based on this comment, we double-checked all citations and references and made sure they were correct and complete. 

 

Reviewer #1

I believe authors have addressed all the major concerns in a reasonable manner; I like that they reanalyzed the data and added new participants. The intro and discussion are much better written now.

Our response: We appreciate your positive feedback. We have indeed made a major revision of the manuscript, and we are pleased that you are satisfied with it.

 

Reviewer #2

Thank you for making changes in the manuscript. I identified some minor issues that should be fixed:

1. Check reference mentioned in line 58 – reference number 6 is missing.

Our response: Thank you for pointing this out. During the revision process, reference #6 was accidentally omitted from the text. It has now been added as reference #2. The text and reference list have been corrected accordingly.

2. Openness to changes should be openness to experiences – please revise it in whole manuscript

Our response: Thank you for this comment. We used the term open to experiences rather than Open to changes throughout the manuscript, including tables and figures.

3. Encore – wrong spelling, it should anchors.

Our response: In line with this comment, the typo was corrected. 

4. Correlations should be presented with 2 decimals.

Our response: We have corrected both the correlation table and the correlation information in the text to show 2 decimals.

---

## [Editor Report · Decision Letter 2]

17 Dec 2021

Dispositional and situational personal features and acute post-collision head and neck pain: Double mediation of pain catastrophizing and pain sensitivity

PONE-D-20-40180R2

Dear Dr. Granot,

We’re pleased to inform you that your manuscript has been judged scientifically suitable for publication and will be formally accepted for publication once it meets all outstanding technical requirements.

Kind regards,

Inmaculada Riquelme

Academic Editor

PLOS ONE
---

## [Editor Report · Acceptance letter]

31 Dec 2021

PONE-D-20-40180R2 

Dispositional and situational personal features and acute post-collision head and neck pain: Double mediation of pain catastrophizing and pain sensitivity 

Dear Dr. Granot:

I'm pleased to inform you that your manuscript has been deemed suitable for publication in PLOS ONE. Congratulations! Your manuscript is now with our production department. 

Kind regards, 

on behalf of

Dr. Inmaculada Riquelme 

Academic Editor

PLOS ONE